# A Conceptual Architecture in Decentralizing Computing, Storage, and Networking Aspect of IoT Infrastructure

Yustus Eko Oktian , Elizabeth Nathania Witanto and Sang-Gon Lee *

College of Software Convergence, Dongseo University, Busan 47011, Korea;
d0185099@kowon.dongseo.ac.kr (Y.E.O.); d0205114@kowon.dongseo.ac.kr (E.N.W.)
* Correspondence: nok60@dongseo.ac.kr

**Abstract:** Since the inception of the Internet of Things (IoT), we have adopted centralized architecture for decades. With the vastly growing number of IoT devices and gateways, this architecture struggles to cope with the high demands of state-of-the-art IoT services, which require scalable and responsive infrastructure. In response, decentralization becomes a considerable interest among IoT adopters. Following a similar trajectory, this paper introduces an IoT architecture re-work that enables three spheres of IoT workflows (i.e., computing, storage, and networking) to be run in a distributed manner. In particular, we employ the blockchain and smart contract to provide a secure computing platform. The distributed storage network maintains the saving of IoT raw data and application data. The software-defined networking (SDN) controllers and SDN switches exist in the architecture to provide connectivity across multiple IoT domains. We envision all of those services in the form of separate yet integrated peer-to-peer (P2P) overlay networks, which IoT actors such as IoT domain owners, IoT users, Internet Service Provider (ISP), and government can cultivate. We also present several IoT workflow examples showing how IoT developers can adapt to this new proposed architecture. Based on the presented workflows, the IoT computing can be performed in a trusted and privacy-preserving manner, the IoT storage can be made robust and verifiable, and finally, we can react to the network events automatically and quickly. Our discussions in this paper can be beneficial for many people ranging from academia, industries, and investors that are interested in the future of IoT in general.

**Keywords:** IoT architecture; distributed system; IoT workflow; blockchain; software-defined networking

## 1. Introduction

Since its inception, the Internet of Things (IoT) has opened vast possibilities to improving our life. Due to the ability to collect more data from IoT devices, we can better assess our environment. Statista predicts that the number of IoT-connected devices will reach 75 billion devices worldwide in 2025 [1]. Machina Research (now acquired by Gartner) also estimates that the IoT market will generate 3 trillion USD revenue in 2025 [2]. Despite all of this growth, IoT still has several issues to be solved in terms of computing, storage, and networking. Most of them exist solely because of the centralized nature of the state-of-the-art IoT architecture.

Most IoT devices are resource-constrained devices that cannot perform complex computations by themselves. As a result, the computation is shifted to a central server that resides in the Cloud, which holds enough processing power to process the IoT data. This processing workflow requires the IoT devices to give up their sensing data to the Cloud for analytics [3]. If the IoT devices collect sensitive information such as healthcare data, then data privacy is broken by design in this centralized architecture.

Before the IoT data analytics begin, the data must be stored somewhere first. Due to the centralized pipeline, the server in the Cloud becomes the suitable candidate for storing the data as that place is where the processing happens. Because all IoT data is stored in a

relatively "single" location, it eases the hackers' attempt to find targets for their attacks. They will be most likely to attack the Cloud environment by stealing the data [4] or disrupt their operations through ransomware [5].

On the other hand, the massive IoT data also brings complexity to the networking sector. The IoT infrastructure may suffer from packet bursts due to the possibility that many IoT devices send their data simultaneously. Moreover, IoT applications (e.g., in the smart disaster prevention application) may require fast responses to IoT events. Therefore, IoT data need to be sent to the Cloud quickly, and the devices must receive feedback as soon as possible. The current centralized networking infrastructure poses severe threats from these scalability issues.

To revolutionize IoT infrastructure, researchers begin to take an interest in the decentralized IoT architecture as an alternative to replace the outdated centralized paradigm (c.f. [6] to get more insights on the trade-offs between centralized and distributed IoT architecture). On the other side, the blockchain, the underlying technology behind the Bitcoin [7], gained popularity recently. Bitcoin proved that complete decentralization (without trusted party intervention) is possible. This news brought blockchain adaptation to many sectors such as IoT [8]. Consequently, it transfers the decentralization hype to the IoT sector as well.

Following the quest to develop a decentralized architecture for IoT, we propose a decentralization strategy throughout three spheres of computing, storage, and networking in this paper. We believe that they are all intertwined in the general-level of IoT workflows. Therefore, they all deserve decentralization re-works. The goal and contribution of this paper are to introduce a possible decentralized architecture for IoT using the building blocks from our previous literature survey. They are mostly a combination of blockchain, distributed storage, and software-defined networking (SDN).

In our conceptual architecture, the government maintains three decentralized, shared services: peer-to-peer (P2P) Computing Overlay, P2P Storage Overlay, and P2P Networking Overlay. The IoT domain owners govern their domain independently, comprising IoT gateway, local SDN switches, and IoT devices. They provision IoT services to IoT users through their IoT gateway. The Internet Service Provider (ISP) provides connectivity across IoT domains through their autonomous system (AS), which comprises ISP server and root SDN switches. Finally, the IoT developers build their applications in the form of decentralized applications (DApps) and deploy them in IoT gateways. The DApps can connect to the provisioned decentralized services to provide various IoT workflows such as computing, storage, and networking.

Through the given IoT workflows, the gateways can train IoT data distributedly without sharing the IoT data with other entities, therefore preserving data privacy. However, the gateways can also share their collected data if they want to by storing it in decentralized storage. The hash of the data will also be stored in the blockchain. The receiving party can then validates the integrity of the shared data by comparing the hash of the received file with the one in the blockchain. Finally, the networking overlay allows the gateways and the ISP servers to collaborate in providing intra- and inter-domain routing across all entities. Overall, our conceptual architecture can serve as an alternative to the existing IoT architectures by bringing decentralization benefits to the IoT infrastructure.

We organize the rest of this paper as follows. Section 2 investigates the related research papers on decentralizing IoT architecture. Section 3 outlines the background and motivation of this paper. We then discuss our proposed architecture from low-level perspective in Section 4, while the high-level perspective is discussed in the following Section 5. Several examples of IoT workflows that IoT developers can perform in our architecture are presented in Section 6. Finally, we discuss limitations and future research directions in Section 7, and concludes in Section 8.

## 2. Related Works

Several kinds of research regarding decentralized computing for IoT exist in the literature. Kolhar et al. [9] construct a decentralized IoT-based biometric face detection for lockdown cities during COVID-19 outbreaks. To fully restrict public movements, they propose three-layer architecture (i.e., application layer, edge layer, and device layer) and then build a deep learning framework on top of it. The framework can perform multi-task cascading to recognize the face of individuals. BlockIoTIntelligence [10] combines blockchain and artificial intelligence (AI) for IoT big data analysis purposes. It comprises three layers of architecture: cloud, fog, and edge; all are integrated using a blockchain network. The given quantitative analysis shows that the proposed architecture produces small overheads compared to similar architecture without blockchain integration. BlockSecIoTNet [11] proposes a decentralized security architecture using SDN, blockchain, and fog servers for a smart city environment. SDN is used for continuous monitoring of IoT traffics to collect necessary data for detection. Meanwhile, the actual attack detection happens in each of the fog nodes. The blockchain is used to deliver a distributed detection while also mitigate the single-point-of-failure problem.

Concept papers for decentralized computing are also available. Ning and Wang [12] propose a future IoT architecture concept in two aspects: Unit IoT and Ubiquitous IoT. The Unit IoT refers to the basic IoT unit with specific special applications, which has a man-like nervous (MLN) model. On the other hand, the Ubiquitous IoT concept refers to an "everything connected, intelligently controlled, and anywhere covered" architecture, a modified and enhanced version of the MLN model. Sarkar et al. [13] propose a layered IoT architecture to ensure interoperability among many IoT devices. The Virtual Object Layer (VOL) is responsible for the virtualization of IoT physical objects or entities. The Composite Virtual Objects (CVO) coordinates those entities in the VOL. Finally, the Service Layer (SL) is responsible for creating and managing IoT services, which abstract the underlying IoT process to the IoT applications.

Aside from computing, decentralized storage has also been investigated in the literature. Narendra et al. [14] propose decentralized cloud-based storage for IoT data. The components of their approach include software-defined storage and mini-data centers. Specifically, they find solutions on where to place the mini-data centers to minimize the latency, ranging from IoT data collection to data migration. ChainSplitter [15] is a hierarchical blockchain storage strategy in industrial IoT (IIoT) to eliminate blockchain storage issues, where all nodes must store all of the data locally. Using ChainSplitter, most of the blocks are stored in the clouds, whereas most recent blocks are stored in the overlay networks in the IIoT network. Two connectors exist in the architecture to bind the cloud storage with the blockchain storage.

The data from the decentralized can then also be used for an efficient data sharing mechanism. BeeKeeper [16] combines blockchain and homomorphic encryption (HE) for threshold IoT service systems. In their systems, users encrypt their data using HE and store them in the blockchain. The servers then can query the encrypted data and perform computation in the cipher space using HE. This way, the data privacy of the users is preserved. Zhu et al. [17] use blockchain to build a decentralized platform for storing and trading IoT data. They acknowledge the limitation that blockchain requires high computing resources. Therefore, they design a novel consensus mechanism involving uneven equilibrium distributions of resources among the participants. The authors then use a Cournot model to optimize the active density factor and Nash equilibrium to balance the number of sensors.

Driven by the motivation to push IoT computations to the edge and the network, Dragon [18] develops a scheme to enable distributed IoT network architecture without connecting to gateways or servers. Dragon can facilitate the users' query for IoT resources or commands and routes it to the intended nodes efficiently, all without any central entity involved in the process. Nour et al. [19] propose a decentralized IoT network architecture based on the named data networking concept. Their proposal includes device and service

networking, communication model, network management, and network naming. They also consider node mobility, hand-off, packet design, and push and pull data service scenarios in their architecture.

In the realm of SDN-based infrastructure for IoT, DistBlackNet [20] combines SDN and Network Function Virtualization (NFV) to form a distributed secure SDN-IoT architecture for smart cities. By using the concept of distributed SDN controllers, the authors separate the whole IoT network into several clusters. Their approach leads to better results in network performances, security, and privacy. Still related to SDN-IoT architecture, DistBlockNet [21] proposes the use of blockchain to secure a distributed SDN for IoT. The authors propose a new Flow Rules insertion scheme using the blockchain, which enables SDN entities to verify the Flow Rules' origin, validate the Flow Rules table, and download the latest Flow Rules table without the need of a trusted intermediary. SoftNet [22] utilizes SDN to manage mobile network architecture so that it becomes 5G-ready. The authors' strategy includes defining the architecture dynamically, decentralized mobility management, distributed data forwarding, and multi-RATS coordination. Based on those ideas, they can design an efficient and scalable network that improves the overall system performance.

Similar to our proposal, the previously mentioned researches are related to distributed IoT architecture. However, we cannot provide a fair comparison in terms of efficiency or usefulness between our works with others as the scope of this paper is only to introduce our conceptual architecture. Therefore, in Table 1 we measure our comparison based on the coverage area. From that table, we can see that other researchers only consider decentralization in one or two aspects from computing, storage, and networking workflow. Meanwhile, we provide all aspects of workflows in our architecture. This trend shows the necessity to provide decentralization in all aspects of IoT to realize a fully decentralized IoT infrastructure.

**Table 1.** The comparison between our proposed architecture and the state-of-the-art based on their decentralization coverage in terms of computing (Comp.), storage (Stor.), and networking (Net.)

| Ref. | Comp. | Stor. | Net. | Description |
|------|-------|-------|------|-------------|
| [9] | ✓ | ✗ | ✗ | Three-layered decentralized deep learning framework for face detection. |
| [10] | ✓ | ✗ | ✗ | Three layers blockchain-based cloud, fog, and edge architecture. |
| [11] | ✓ | ✗ | ✗ | Cyberattacks detection with SDN, blockchain, and fog servers. |
| [12] | ✓ | ✓ | ✗ | A future IoT architecture comprising Unit IoT and Ubiquitous IoT. |
| [13] | ✓ | ✓ | ✗ | A layered IoT architecture considering IoT devices' interoperability. |
| [14] | ✗ | ✓ | ✗ | A software-defined storage and mini-data centers architecture. |
| [15] | ✗ | ✓ | ✗ | Hierarchical blockchain storage strategy in industrial IoT. |
| [16] | ✗ | ✓ | ✗ | Blockchain and homomorphic encryption for data sharing. |
| [17] | ✗ | ✓ | ✗ | Blockchain-based data storage and trading for IoT sensors. |
| [18] | ✗ | ✗ | ✓ | A distributed network scheme for gateway-less architecture. |
| [19] | ✗ | ✗ | ✓ | A decentralized IoT architecture based on the named data networking. |
| [20] | ✗ | ✗ | ✓ | SDN and NFV for secure IoT architecture in smart cities. |
| [21] | ✗ | ✗ | ✓ | A combintaoin of blockchain and SDN to secure distributed IoT. |
| [22] | ✗ | ✗ | ✓ | SDN-enabled mobile network architecture for 5G. |
| Ours | ✓ | ✓ | ✓ | A conceptual integration of blockchain, distributed storage, and SDN architecture for IoT computing, storage, and networking. |

## 3. Preliminaries

This section mentions several background and motivations of our concept paper in decentralizing the computing, storage, and networking aspect of IoT architecture.

### 3.1. Decentralizing IoT Computing

The essential issue in decentralizing IoT computation is trust. Specifically, it is challenging to trust operations given by others, especially in IoT cases where the number of

gateways or edge servers capable of doing computation is large. It is easier to verify a single server rather than, let us say, thousands of gateways. As a result, we consider using blockchain as our trusted platform to provide computation services.

Entities in the blockchain send messages to one another in the form of transactions. In doing so, they must provide digital signatures for each transmitted transaction. Other nodes pick the transactions and then construct a block, which contains a list of transactions. They hash the constructed block while including the hash of the previous block, forming a chain-of-hashes. The blockchain consensus algorithm makes sure that all nodes maintain the same data entry at all times. Together, they all provide guarantees that all data stored in the blockchain network becomes hard-to-tamper (c.f., Bitcoin whitepaper [7] to get more details about how the blockchain works).

The tamperproof guarantee also enables the blockchain to run secure codes distributedly through a "smart contract" [23]. To propose a code execution in the smart contract, entities must form a transaction to the smart contract, similar to sending transactions in the general blockchain. The code is then executed simultaneously in all blockchain nodes when they reach consensus (c.f., Ethereum whitepaper [24] for more details regarding the smart contract executions). Using smart contracts, we can provide a trustable execution platform, which can be run in a decentralized manner.

Ideally, all of the blockchain that supports smart contracts are eligible for our architecture. However, we assume Ethereum [24] for our P2P computing services in this architecture. Ethereum is the second-largest cryptocurrency based on its market cap, ensuring its usability. Ethereum has rich smart contract features and also receives comprehensive community supports. Therefore, many decentralized services are built on top of Ethereum.

### 3.2. Decentralizing IoT Storage

Considering the high number of IoT devices, it is expected that IoT generates many data during its daily operations. We can theoretically store those data in the blockchain; however, it is not practical. Storing many files in the blockchain is very costly in terms of money and spaces. In Ethereum [24], the more data stored in the smart contract, the higher the price that the sender has to pay. Furthermore, because of how replication works in the blockchain, sending many files to the blockchain means that all nodes must store them in their internal storage. This behavior may result in unnecessary redundancy that makes the storage system becomes inefficient.

We decided to use a separate distributed storage in our architecture in search of solutions for this problem. Compared to the blockchain, this storage system is not tamperproof; therefore, it is not entirely secure. However, it is very efficient as the data stored is not replicated to all nodes. The data can be split or duplicated to several nodes if necessary to provide robustness (c.f., InterPlenary Storage System (IPFS) whitepaper [25] to get insights on one of the examples of how decentralized storage works).

Many decentralized storage systems exist in the literature, and theoretically developers can use any storage system as they see fit in their architecture. However, we assume the use of IPFS [25] for our P2P storage services in this architecture. When we store arbitrary data in IPFS, the network returns the hash of the data. This hash becomes the data identifier, and anyone with access to this hash can query the data contents from the network. Our decision in picking IPFS is because of this convenience.

### 3.3. Decentralizing IoT Networking

Our network has been decentralized and vendor-specific for a very long time. Managing the network in this distributed manner is very challenging and error-prone because of human error [26], especially for IoT networks, where the data volume traffics and the number of nodes is very high, resulting in many configuration requirements in switches and routers. Moreover, some traffic needs to be processed quickly. For example, a smart healthcare system needs a high-speed response network to save someone's life.

A new networking paradigm called software-defined networking (SDN) arises to solve those problems. SDN takes out the software or the "brain" of networking devices (e.g., switches and routers) and aggregates them into a new server called "SDN controller" [27]. This controller dictates the underlying switches and routers' behavior in daily networking tasks such as managing traffics [28] or enforcing network security [29]. We can react to dynamic network events more quickly through this novel centralized management approach than the previous decentralized method. The network configuration can also be made automatic by the SDN controller.

Ironically, the pilot SDN controller has issues with its robustness and scalability (e.g., single-point-of-failure problem) because of its centralized architecture. Researchers then take back the decentralized approach by fusing SDN with the distributed system concept (c.f., ONOS [30], one of the famous distributed SDN controllers). In particular, several SDN controllers now exist in the network, and they must share their local network view. All controllers then apply a consensus algorithm and form an aggregated global network view. This approach resembles how the blockchain works; therefore, we can integrate this network with blockchain to augment its integrity.

There are many alternatives for Southbound protocol in SDN. However, throughout the rest of this paper, we pick OpenFlow [27] to employ P2P networking services in our IoT architecture. OpenFlow is the most famous SDN protocol, and many SDN controllers support it. By choosing OpenFlow, we indirectly ease the future implementation of this architectural design.

## 4. Low-Level Architecture

From a low-level perspective, our proposed architecture can be divided into four major physical or logical hardware components: the overlay networks, the IoT actors, the IoT domains, and the autonomous systems. We summarize them in Figure 1 and elaborate as follows.

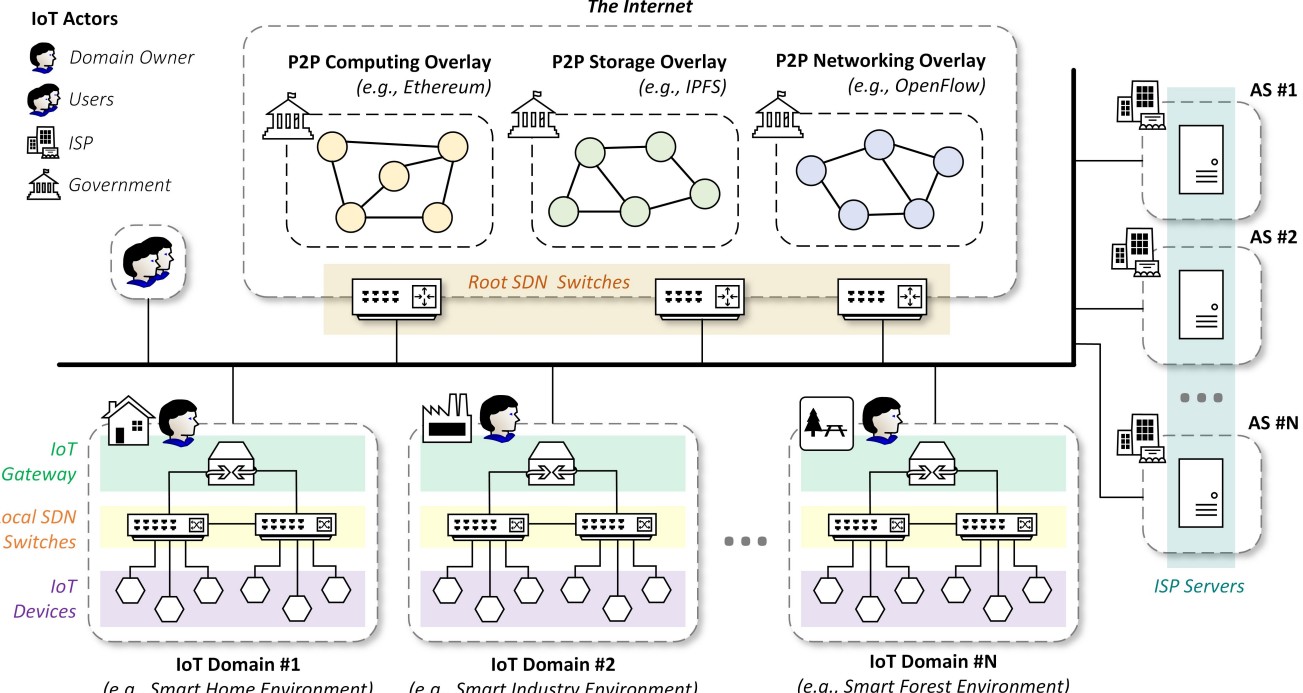

**Figure 1.** Our conceptual decentralized IoT architecture from a low-level perspective. The government manages three overlay networks, which host shared decentralized services. The domain owners maintain the IoT domains and supply IoT resources to IoT users. Finally, the Internet Service Provider (ISP) provides connectivity to others through their autonomous system (AS).

### 4.1. Overlay Networks

We employ three peer-to-peer (P2P) decentralized networks in our IoT architecture to provide computing, storage, and networking services. They reside as overlay networks inside the Internet. As a result, anyone with access to the Internet is also connected to those services.

*P2P Computing Overlay*: The first overlay network is responsible for providing shared computing services such as identity management, distributed data training, and trusted data sharing. Refer to Section 5.1.1 for examples of such services. We employ smart contracts that Ethereum blockchain [24] provision as our P2P computing platform.

*P2P Storage Overlay*: The second overlay network provides storage for all IoT entities. In particular, they can store IoT data, neural network models, or application data in the network. Refer to Section 5.1.2 for examples of such services. We use IPFS [25] to store arbitrary data distributedly across IoT entities.

*P2P Networking Overlay*: The final overlay network manages networking-related services such as maintaining intra- or inter-domain routing tables, access control rules (firewall rules), and gathering network statistics. Refer to Section 5.1.3 for examples of such services. We employ SDN and the OpenFlow protocol [27] to provision this service.

### 4.2. IoT Actors

We define IoT actors as those that exist in our architecture as humans or organizations. Therefore, IoT devices, gateways, switches, and servers are not actors even though they are owned or managed by a person or a company.

*IoT domain owners*: IoT practitioners from multiple sectors, such as smart manufacturing factories, smart home companies, and government smart city management, may exist in our architecture. They all follow the principle of "bring your own domains" into the system. More specifically, they can arbitrarily join our platform and form IoT domains by deploying their IoT devices, local SDN switches, and IoT gateways in their administrative area and connect them to our decentralized computing, storage, and networking services (i.e., the overlay networks). As a result, those practitioners become the domain owners responsible for maintaining their domains independently.

*IoT users*: Contrary to the domain owners, which manage the domain, IoT users consume the resources from the IoT domains. Due to the lack of a centralized server (or Cloud) in our architecture, the users must ask for IoT services through the domain owners' gateways as the new "servers". The gateways provision multiple services in the form of decentralized applications (DApps).

*Trusted government and law enforcements*: The previously mentioned overlay networks in Section 4.1 are shared among all actors. A trustable law must exist to regulate the network usage such that any malicious actions will be held accountable. This regulation ensures the security and fairness of the shared services, which can be realized by two actors: the government and law enforcement. More specifically, the government designs fair and secure usage policies, and law enforcement drives them into action.

*Internet Service Providers (ISP)*: This final actor provides communication services among IoT domains. With this backbone communication, IoT devices from one domain can interact with others reside in different domains. Like the concept of IoT domains, ISPs also maintain their administrative domain, also known as autonomous systems (AS). IoT traffics from a particular domain may traverse through multiple ASs before reaching their destinations, which can be subject to the policies of several different ISPs.

### 4.3. IoT Domains

The domain owners form an arbitrary IoT domain, which comprises IoT devices, local SDN switches, and IoT gateways.

*IoT devices*: IoT devices can be any devices (e.g., sensor or actuators) [31], and their detail specification varies depending on the actual use cases. For example, devices in a "smart forest or smart agriculture" environment, which are most likely to reside in rural

areas, may have lower computing, storage, and networking resources than those in the "smart factory or smart home" sector. Our proposed architecture is general enough to cope with various use cases.

*Local SDN switches*: Sets of local SDN switches exist in the domain to control the intra-domain and inter-domain traffics [28]. The administrator may separate their roles into logical switches and routers, with switches control the intra-domain traffics and routers control inter-domain traffics. This way, IoT devices can all be connected and also attached to entities outside their domain. All local SDN switches correlate to the local SDN controller in the administrative IoT gateway for that domain.

*IoT gateways*: IoT gateway is the entry point, and manager for the IoT domains, in which all decisions on what traffics or operations performed in the domain is resolved. The domain owner must administer this gateway. As a result, all actions made inside the gateway are subject to the domain owner's regulation.

## 4.4. Autonomous Systems

Autonomous Systems (AS) are a big group of networks that form the Internet [32]. The ISPs control each of these ASs, and they are responsible for providing routing for all traffics across the globe. In our architecture, those ASs are equipped with our decentralized services, enabling multiple ISPs from different ASs to collaborate efficiently.

*Root SDN switches*: Sets of root SDN switches exist in the AS to control inter-domain traffics. The IoT traffics from a particular domain to other domains will most likely go through root SDN switches in the process, especially if the domains are separated across far distance. The routing of those traffics usually maintained using Border Gateway Protocol (BGP). However, the vanilla BGP is prone to error [26]; therefore, automatic configuration using SDN is preferred.

*ISP servers*: The ISP server is the manager for the AS, which controls the routing of inter-domain traffics through its root SDN controller. A specific ISP governs a particular AS, and thus the routing decision is subject to that ISP regulation. Through collaboration with other ASs, ISPs can provide routing across the globe, thus forming the Internet.

## 5. High-Level Architecture

From a high-level perspective, our proposed architecture comprises many software components scattered in overlay networks, IoT devices, SDN switches, IoT gateways, and ISP servers. We summarize them in Figure 2 and elaborate as follows.

### 5.1. Software in Overlay Networks

The overlay networks hold three "physically separated but logically integrated" shared services: computing, storage, and networking services. There should be no boundary on what kind of services that one can offer. Therefore, to maintain its feasibility, we look for several kinds of research in the literature discussing potential computing, storage, and networking use cases. Then, we pick several of them as the primary shared services in our architecture, which we describe as follows.

#### 5.1.1. P2P Computing Overlay

*Identity Management*: All entities in our architecture must be identifiable using a unique identifier (UID). For this purpose, we employ Ethereum addresses as our UID, where all nodes are equipped with a unique Ethereum address. They can use their address to sign a message or to build a secure channel among entities [33].

*Training Aggrement*: When the gateways perform training on IoT data using a Federated Learning (FL) algorithm [34], they must aggregate the submitted models from other gateways, which they do not trust by default. The presence of smart contracts in the computing overlay can help in building trust among gateways. The smart contract can act as an FL organizer for all gateways, which governs the FL process according to the initial training agreement [35].

*Data Sharing Agreement*: When a party wants to share data with others, they can consult the computing overlay to facilitate a trustable data sharing agreement using the smart contract. The sharer must anchor the hash of their data to the blockchain. Then, the receiver can verify the shared data by comparing the hash of the received data with the one previously stored in the blockchain [36].

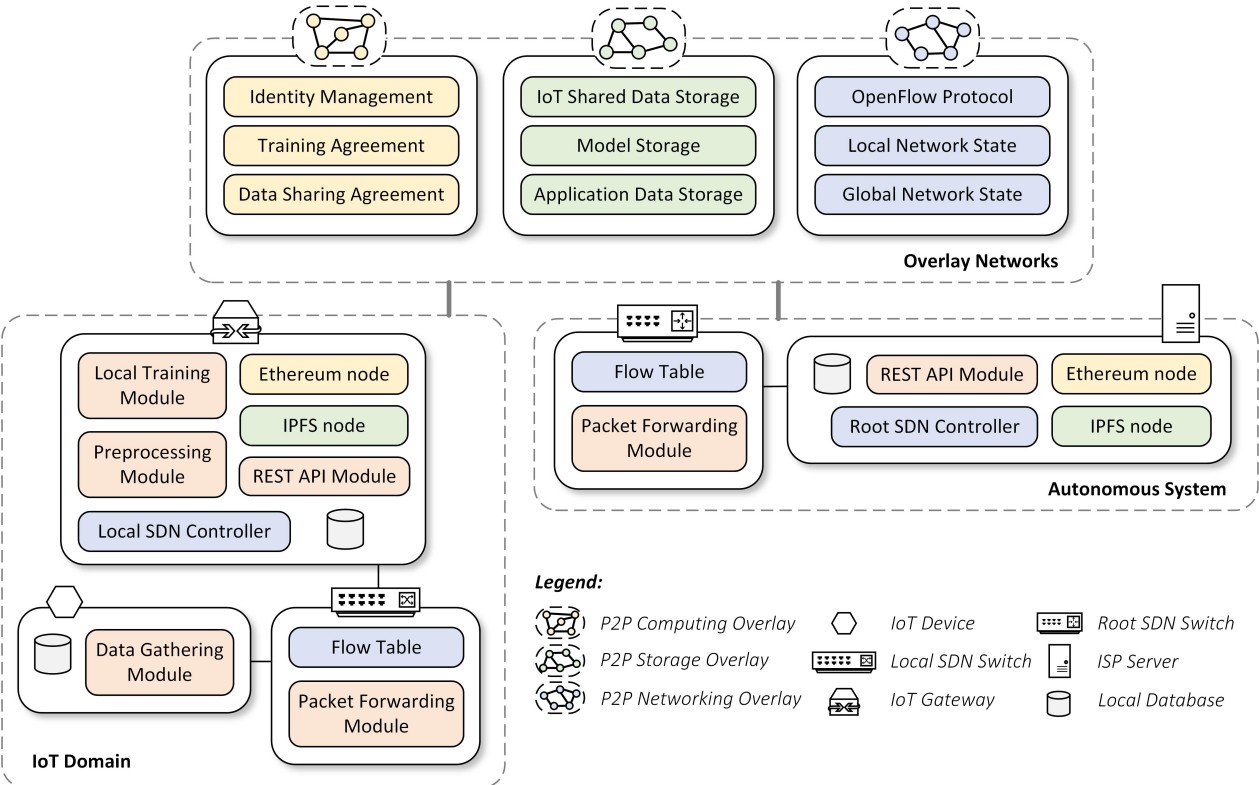

**Figure 2.** Our conceptual decentralized IoT architecture from a high-level perspective shows the necessary software components each entity needs. Red color indicates local processings while yellow, green, and blue colors are related to P2P Computing, Storage, and Networking Overlay, respectively.

### 5.1.2. P2P Storage Overlay

*IoT Shared Data Storage*: The storage overlay can be used for IoT data sharing [37]. IoT users or domain owners can store their IoT data in the storage overlay. They can then give this stored data to others by sharing the location (e.g., URL) to download the data.

*Model Storage*: During the training of IoT data using FL, the IoT gateways (i.e., acts as trainers in FL) get the global model from the storage overlay. After local training, the gateways upload the trained global model back to the storage overlay. Therefore, the storage overlay is used as a mediator to share the global models in FL [35].

*Application Data Storage*: The storage overlay can also be used to share application data [38]. A particular application can share its data through the service to offer interoperability among IoT applications.

### 5.1.3. P2P Networking Overlay

*OpenFlow Protocol*: The root or local SDN controllers can use the networking overlay to manage the underlying SDN switches through OpenFlow protocol [27]. In particular, the controllers can gather some network statistics such as the number of connected devices, the current traffic rate, and traffic type. They can also install Flow Rules (i.e., a forwarding path) to the switches using this protocol.

*Local Network State*: The SDN controllers can form a local network state using the gathered network statistics obtained from the SDN switches. This network state may

produce valuable insights on the local network, such as the number of connected devices, possible congested ports or paths, and an intrusion attempt by attackers [39]. The controller can then store this network state in the networking overlay for management purposes.

*Global Network State*: In the presence of multiple SDN controllers, such as the one in our architecture, the controllers must collaborate to provide inter-domain routing among SDN domains. To do so, they need to share their local network state. The networking overlay can mediate the transfers of local network states to all controllers. Once the controllers receive all network state updates, they can aggregate the updates and form a global network state [39].

### 5.2. Software in IoT Devices

The IoT device has a *Data Gathering Module*, which is responsible for collecting IoT data from its sensors. We encourage the devices to perform only minimal IoT data gathering procedures without any processing involved for efficiency reasons. The gathered data can be stored temporarily in the *Local Database* before transmitting it to the IoT gateway for further processing.

### 5.3. Software in SDN Switches

The primary role of switches is to forward packets. Consequently, all switches in our architecture are equipped with *Packet Forwarding Module*. They also have a *Flow Table*, a module that stores Flow Rules (i.e., the forwarding logic or routing tables) obtained from the SDN controllers. The switches inside the IoT domain (i.e., local SDN switches) query this information from its corresponding local SDN controller (i.e., resides in IoT gateway). Meanwhile, switches in the autonomous systems (i.e., root SDN switches) get that information from the root SDN controller (i.e., resides in the ISP server). The *Packet Forwarding Module* translates the logics in the *Flow Table* into forwarding paths in the application-specific integrated circuit (ASIC) hardware.

### 5.4. Software in IoT Gateways

The domain administrator manages their domain through a centralized IoT gateway, which comprises several software modules.

1. *Ethereum node*: The gateway is a suitable candidate to become a blockchain node due to its computing resource, in case a power-consuming consensus algorithm such as Proof-of-Work (PoW) [7] is used. The gateway also serves additional roles as transaction relayers. Specifically, it has an open channel (e.g., REST API endpoints) for IoT devices to submit transactions to the blockchain through it.
2. *IPFS node*: With its higher storage resource than IoT devices, the gateway can serve as an IPFS node. Together with other gateways, it maintains permanent data stored in the IPFS network and provides data access to other entities when needed.
3. *Local SDN Controllers*: This module controls the underlying SDN switches in the domain and provides networking services through installed SDN applications (e.g., traffic engineering [28] or network security [29]). The gateway maintains a one-to-one mapping between itself and the switches. As a result, the controller in a particular domain cannot take over other switches in different domains. This action is intentional as we assume that a single administrative entity maintains one IoT domain.
4. *Preprocessing Module*: This preprocessing module handles the submitted IoT data from IoT devices and provides micro-processings such as simple anomaly data filtering or data classification [40]. Note that these IoT data will not be transferred to a server as opposed to centralized IoT architecture.
5. *Local training Module*: This module enforces an FL algorithm [34] to enable local training on the preprocessed IoT data. With this FL terminology, IoT gateways from multiple domains may collaboratively train a global model without sharing their private data. In each training iterations, they must share their trained model parameters with others and later aggregate them to form an updated and more accurate global model.

6. *Local Database*: A local database exists in the gateway to provide temporary data storage for ephemeral data (e.g., temporary logs or pointers used for data processing). The gateway deletes the data when it becomes unnecessary anymore. Alternatively, the gateway can persist this data in IPFS or blockchain network when necessary before deleting it.

7. *REST API Module*: The gateway provides IoT services to IoT users from open REST API endpoints. IoT devices can also use those endpoints to submit their IoT data or to submit their blockchain transactions. Recall that the gateway piggybacks all the burden from IoT devices because they have constrained resources. Furthermore, IoT developers leverage this API to deploy SDN applications in the IoT-domain level area. This module is versatile, and it can serve multiple distinct clients simultaneously.

*5.5. Software in ISP Servers*

ISP manages their respective autonomous systems independently from one another through their ISP servers. Those servers have similar software components as IoT gateways. However, they are repurposed for maintaining network communication instead.

1. *Ethereum node*: The server must become an Ethereum node to join the computing overlay and collaborate with other ISPs in negotiating Service Level Agreement (SLA) to route the traffics across IoT domains.

2. *IPFS node*: The server becomes an IPFS node, and together with other ISPs, they share the burden to store permanent networking-related files (e.g., routing information) in the storage overlay.

3. *Root SDN Controller*: This SDN controller governs the underlying Root SDN switches in the respective autonomous systems. The controller also connects to the networking overlay to perform its daily jobs, such as governing the switches and collaborating with other ISPs to provide forwarding paths across IoT domains.

4. *Local Database*: The server has a local database to store temporary data during its operations. For example, maintaining networking logs, saving the history of network view states for backups.

5. *REST API Module*: The ISP opens REST API endpoints to supply networking services to others. Through this API, developers can create SDN applications to be implemented in the AS-level area. The domain owners can also use this module to manage their domain-related networking information.

## 6. IoT Workflows in Our Architecture

Using all of the previously mentioned resources in low-level and high-level architecture, IoT developers can build decentralized applications (DApp) and deploy them in IoT gateways. In theory, there should be no boundary of application use cases. However, we only show some basic DApp examples here, involving storage [38], computing [35], and networking [28] workflows that may exist in general IoT application workflows. We present the generalization of those mentioned reference papers readjusted to match our architecture, which we describe further as follows.

*6.1. Computing Workflows*

During general IoT workflows, users may need to gain insights into their IoT data; however, instead of IoT devices sending private data to the Cloud for analytics as opposed to a centralized workflow. The IoT devices send data to the IoT gateway, which then trains the data distributedly. They can do so by leveraging the concept of FL [34] implemented as smart contract in our P2P Computing Overlay. Figure 3 summarize this process, and we elaborate as follows.

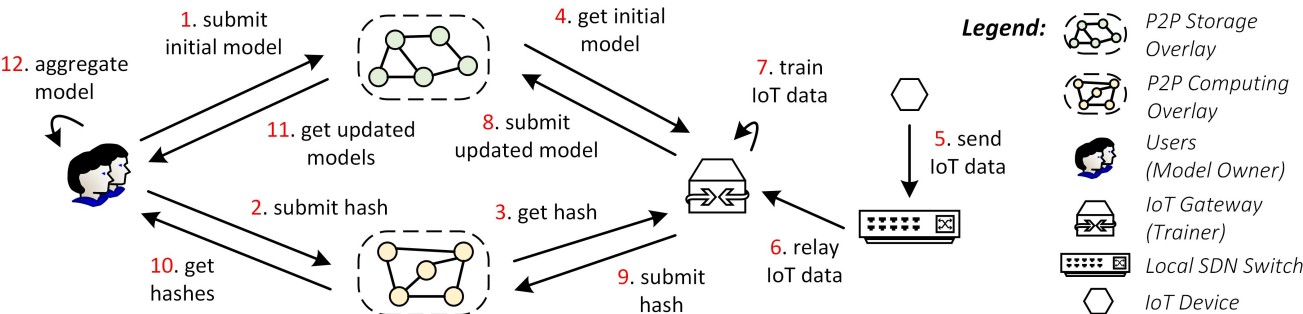

**Figure 3.** Training the IoT data collaboratively using Federated Learning, highlighting a possible computing workflow in our architecture.

First, the IoT users initiate a global model with random parameters and then submit it to the IPFS (Step 1). The users then create a training job (including the training objective) in the smart contract and submit the IPFS hash, pointing where to download the initial global models (Step 2). At any given time, any domain owner (represented by his IoT gateway) can join the training job. After joining, he/she gets the global model's IPFS hash (Step 3) and then downloads the model (Step 4).

During IoT operations, IoT devices send their data to the IoT gateway (Step 5), which is relayed by the local SDN switches inside the domain (Step 6). The gateway then trains the previously downloaded global model using the collected IoT data (Step 7). The gateway trains the model for several epochs until it reaches the required accuracy stated in the training job. Once the training completes, the gateway stores the updated model to the IPFS (Step 8) and submits the IPFS hash to the smart contract (Step 9).

Multiple gateways may participate in the training job. They all submit their training result to the IPFS and smart contract. After the training round finishes, the users collect all IPFS hashes of the updated models from the smart contract (Step 10). Using these hashes, they download the models (Step 11) and aggregate them to get a more accurate model (Step 12). This workflow may repeat for several rounds until the training objectives are achieved. The newly updated global models will now become the initial global model for the next round.

*Benefits*: Committing to this computing workflow improves the privacy and robustness of the system. Because IoT gateway trains the IoT data locally, the data never leaves the IoT domain. Therefore, only IoT devices and gateway know the data; the data privacy is preserved. IoT gateway also collaboratively trains the local model with other gateways and produces a more accurate global model by aggregating all local updates from gateways. Due to this decentralized nature, the training can be continuously performed even when a particular gateway fails when crashing.

### 6.2. Storage Workflows

Before IoT data can be processed, they need to be stored somewhere safe. Instead of storing the data in the centralized server, IoT devices (through the IoT gateway) save the data in the P2P Storage Overlay. Figure 4 summarizes this process, and we elaborate as follows.

During IoT operations, the IoT device generates IoT data and sends them to the IoT gateway through a local SDN switch (Step 1–2). The gateway then submits the data to the IPFS as our P2P Storage Overlay (Step 3) and receives an IPFS hash from this process (Step 4). It then uploads this hash to the smart contract (Step 5). The hash will be stored upon submission, and the gateway gets the transaction (tx) hash as proof of submission. The gateway may perform this storing mechanism several times whenever it receives IoT data from devices in their domain.

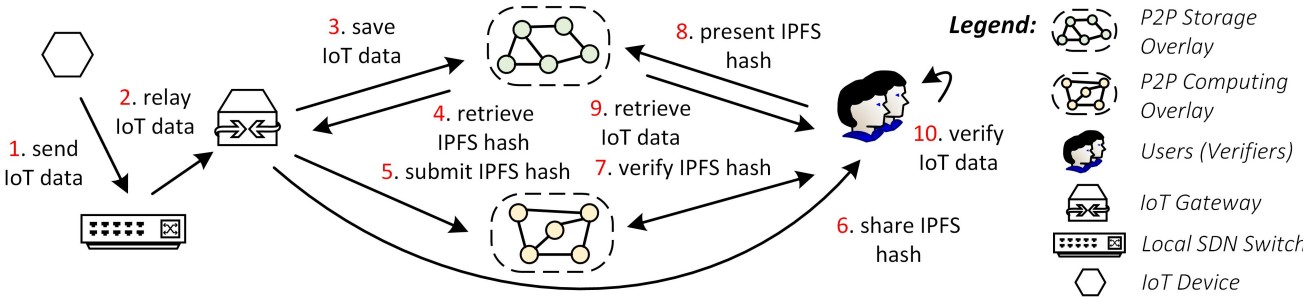

**Figure 4.** Storing and verifying IoT data, highlighting a possible storage workflow in our architecture.

When the IoT data need to be shared or audited, the gateway can disclose the associated IPFS hash to verifiers (Step 6). In this example, we use IoT users as verifiers. Upon receiving the hash, the users validate the IPFS hash to the smart contract (Step 7). First, they must make sure that the miners already validate the sender's tx. Second, they check that the IPFS hash exists in the smart contract. If both verifications are valid, the users can get their IoT data by presenting the IPFS hash to the P2P Storage Overlay (Step 8–9). Finally, the users can use or audit the IoT data (Step 10).

Note that the data stored in the IPFS is not encrypted in this example. However, the gateway can add any encryption algorithm that they want to protect the data confidentiality.

*Benefits*: In this mentioned workflow, the gateway can store and share its data distributedly through the IPFS. Because we store the data in a decentralized manner, they become robust to failure or censorship. Moreover, the smart contract preserves the integrity of the submitted IPFS hash. Any malicious tampering of the data can be easily detected as the modification will change the IPFS hash, and it will not match the previously stored hash in the smart contract.

### 6.3. Networking Workflows

IoT entities transfer IoT data and commands to one another daily. In this final use case, we show a networking workflow of delivering cross-domain IoT data packets. For simplicity, we only assume a round-trip packet delivery, as shown in Figure 5. Our P2P Networking Overlay (i.e., based on Openflow [27]) leads the transmission of this packet from its source to destination and vice versa.

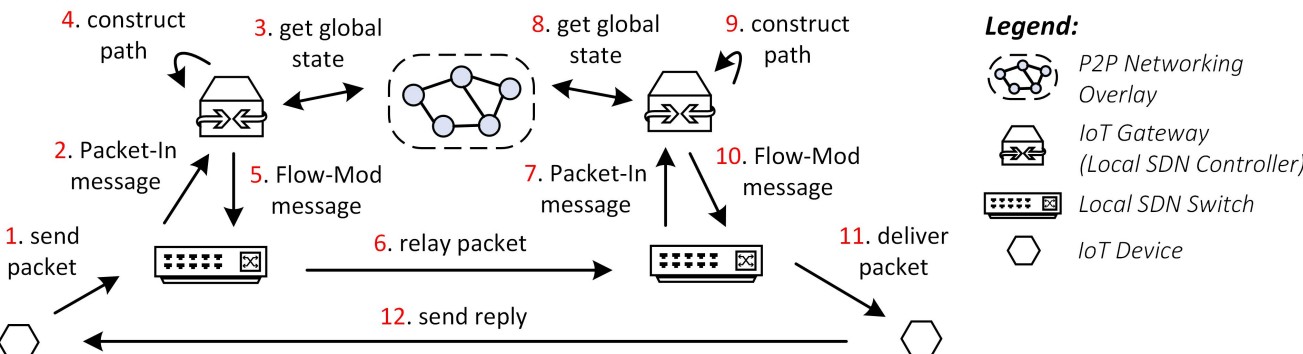

**Figure 5.** Delivering IoT data traffics across IoT domains, highlighting a possible networking workflow in our architecture.

The IoT device *A* crafts an IoT data packet for device *B* and sends it to the SDN switch (Step 1). Assuming that *A* never sends packets to *B* before, the switch sends Packet-In messages to the SDN controller, which resides in the IoT gateway (Step 2). The controller query for an up-to-date global network state from the P2P Networking Overlay (Step 3). Using the newly collected information, the controller then calculates forwarding paths for this packet (Step 4) and installs them in the switch using the Flow-Mod message (Step 5).

Following the previous Flow-Mod message instructions, the switch relays the packet to the destination domain (Step 6). The corresponding SDN switch retrieves this packet and sends Packet-In messages to its SDN controller (Step 7). Similar to what happens in the source domain, the controller gathers an up-to-date global network (Step 8) before constructing forwarding paths (Step 9). The controller then sends a Flow-Mod message to the switch and installs the forwarding paths (Step 10). The switch then transmits the packet to its destination (Step 11).

The IoT device *B* sends a reply for device *A* and delivers the packet to the switch. However, because the forwarding path has been previously installed in Steps 10 and 5, the switches can directly relay the packet to its destination (i.e., device *A*) without contacting the SDN controllers. Therefore, the reply message takes lesser time to deliver compared to the initial message.

Note that the packet delivery across the domain may involve root SDN switches and controllers. However, we omit those steps for simplicity, as they are generally similar to Steps 2–5, in which the Packet-In and Flow-Mod messages are generated in sequence.

*Benefits*: The P2P Networking Overlay is used in this mentioned workflow to maintain an up-to-date global network state of SDN controllers from multiple IoT domains. The overlay also facilitates the OpenFlow message transmission between the SDN controllers and the SDN switches. Therefore, the overlay network plays a crucial role in managing the delivery of IoT traffics efficiently.

## 7. Discussion

This paper's immediate future work will be its implementation and evaluation because we write this paper as a concept paper. We design the architecture based on our literature surveys on emerging technologies such as blockchain, decentralized storage, and software-defined networking. Previous researches discuss implementations of those technologies, and many are available as open-source projects. Therefore, given enough time, we argue that our design should be feasible to implement. However, we project several challenges and issues remain after implementation, which may hinder the applications in the production setting. We mention some of them in the following paragraphs.

In terms of computing, the smart contract's execution is expensive to perform in a public network (i.e., permission-less blockchain) because users must pay the miners to perform the PoW algorithm to secure the network. For the same security reason, the blockchain must also throttle the throughput to ensure consistency across all blockchain nodes. This decision limits the throughput of the network to only perform about 15 transactions per second for the Ethereum case. Moreover, PoW is also power-hungry, and it is not suitable for the environment [41].

As an alternative, adopters can deploy a private network (i.e., permissioned blockchain) to boost the system's scalability. For example, Quorum [42], a modified Ethereum blockchain tailored for enterprises, uses Byzantine Fault Tolerance (BFT) consensus protocol to enable thousands of transactions per second throughput. However, BFT-variant consensus only allows a limited number of nodes. As a result, there is a trade-off between a high throughput or a high number of nodes [43].

Future research directions for our P2P Computing Overlay should enhance the blockchain network's performance, especially for IoT cases, where the data traffic volume is high and may come from many sources of IoT devices. The current state of blockchain performance will cause a bottleneck in the system.

In terms of storage, the file stored in the decentralized storage is hard-to-trace and hard-to-delete. It is very challenging to determine who is the origin or owner of a particular stored data, even though this feature may still be proven helpful in some scenarios (e.g., the data cannot be censored by an authoritarian government). The saved data also become hard-to-delete because of the duplications made to ensure high availability. As a result, when the users want to remove the data, they must find the stored data locations before deleting it.

To solve the hard-to-trace issue, we can make rules that all stored data must be signed and logged in the blockchain, as shown in our Storage Workflow example in Section 6.2. This way, we can quickly determine who uploads the given data to the P2P Storage Overlay. Meanwhile, solving the hard-to-delete problem is more complicated. It may involve finding an efficient algorithm for deduplication. However, if we de-duplicate the data, it becomes less compromised to failure if the node saving that data fails.

Further research must be conducted to find solutions for the trade-off to ensure data stability, thus ensuring high availability and easy-to-delete, especially for IoT cases, where most of the IoT data is ephemeral data. We may not need the raw data after we obtain micro- or macro-processing results. Furthermore, querying data from decentralized storage is also slower compared to centralized storage. Hence, enhancing the query performance is also an essential requirement.

In terms of networking, SDN controllers and the underlying SDN switches can collaborate through the OpenFlow protocol. However, there is no standardized protocol that governs the collaboration among SDN controllers. This absence may pose problems in our P2P Networking Overlay because ISP from one AS may have different opinions in handling the network than the one in other ASs. Consequently, they may find it difficult to reach agreements and continue to maintain the network moving forward.

Future protocol design to solve the ISP interoperability issue is necessary, especially for IoT network case, which usually covers a wide-level management area comprises of multiple networks at city-level, national-level, or international-level. This vast area coverage means more frequent collaborations between ASs are required.

## 8. Conclusions

This paper introduced a conceptual decentralized IoT architecture throughout three IoT workflows: computing, storage, and networking. The Ethereum smart contract facilitated secure computation for IoT entities in the form of P2P Computing Overlay. The IPFS, as our P2P Storage Overlay, maintained the saving of IoT data, FL model, or application data distributedly. SDN controllers and SDN switches, as P2P Networking Overlay, were leveraged to govern the intra-domain and inter-domain communication. IoT developers built their customized applications using those shared overlays and deployed them in IoT gateways. By committing to our architecture, the IoT computation could be made private without leaking the IoT data while maintaining robust IoT data storage and reactive IoT networking. Because we only proposed our architecture design in this paper, its implementation and evaluation became our immediate future works. Meanwhile, several limitations and future research directions for realizing our proposal were also discussed in this paper.

**Author Contributions:** Conceptualization, Y.E.O.; methodology, Y.E.O.; validation, Y.E.O. and E.N.W.; investigation, Y.E.O. and E.N.W.; resources, Y.E.O. and E.N.W.; writing—original draft preparation, Y.E.O.; writing—review and editing, Y.E.O., E.N.W., and S.-G.L.; visualization, Y.E.O.; project administration, S.-G.L.; supervision, S.-G.L.; project administration, S.-G.L.; funding acquisition, S.-G.L. All authors have read and agreed to the published version of the manuscript.

**Funding:** This work was supported by Basic Science Research Program through the National Research Foundation of Korea (NRF) funded by the Ministry of Education (Grant Number: 2018R1D1A 1B07047601).

**Institutional Review Board Statement:** Not applicable.

**Informed Consent Statement:** Not applicable.

**Data Availability Statement:** Not applicable.

**Acknowledgments:** We would like to thank the anonymous reviewers for their comments and suggestions that help us improve the paper.

**Conflicts of Interest:** The authors declare no conflict of interest.

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
