# Peer review of "A Conceptual Architecture in Decentralizing Computing, Storage, and Networking Aspect of IoT Infrastructure"

_2624-831X, doi:10.3390/iot2020011_

Round 1
Reviewer 1 Report
General Comments
This paper is Towards Decentralizing Computing, Storage, and Networking Aspect of Internet of Things Architecture. The area is interesting and the paper is not bad.
However, there are some suggestions to the authors for improving the work:
Specific Comments
1) Please cross mark used in Table 1 when field value is empty.
2) Please add more related papers in "Related works" section.
3) Please check the grammer mistake.
Author Response
Comment #1
General Comments
This paper is Towards Decentralizing Computing, Storage, and Networking Aspect of Internet of Things Architecture. The area is interesting and the paper is not bad.
However, there are some suggestions to the authors for improving the work:
Response #1
We thank the reviewers for their useful comments.
Comment #2
Specific Comments
1) Please cross mark used in Table 1 when field value is empty.
Response #2
We have added cross marks in Table 1.
Comment #3
2) Please add more related papers in "Related works" section.
Response #3
We have added more related papers (c.f. Section 2)
Comment #4
3) Please check the grammer mistake.
Response #4
We have re-checked the grammar in our manuscript.
Reviewer 2 Report
- This reviewer believes storage is not very important and may not be required in the title. Only computing and Networking is ok!
- Highlights on the major contributions with emphasis seems missing in the abstract.
- Intro 1st para: Machina Research announces its acquisition by Gartner (the world’s leading information technology research and advisory company) long back.
- Page-2 Motivations to this work requires substantial improvements with clarity of flows and arguments since it is redesigning the architecture and challenging both the layering centralized approach in some sense. This is definitely a delicate balancing act.
- Literature Page-3: There are a few important works elaborating the decentralization even in the case of mobile IoT networks. It seems that some relevant and important works are not covered here.
- Section 3.4 requires further elaboration with literature and benchmarking.
- Overlay networks: It requires strong justification with valuable references from literature to support such ideas.
- Storage workflows: This reviewer is not fully convinced with the storage workflows. Perhaps we can move it as a subsection of the network. I know this is indeed a major change that requires authors to polish the manuscript from top to bottom to tone down the storage aspects and highlight networking and computing aspects.
- Computing workflows: How it works is explained clearly however, its significance and importance, challenges, impeding issues etc have not been covered thoroughly, which is indeed what the readers and the research community will be looking for in such papers.
- Discussion: We need to bring out some limitations, shortcomings and complexity of handing such architecture in future IoT networks. Any supporting preliminary results, if viable to produce helps to justify such development and findings in the paper. BTW, potential future directions and challenges to be tackled with plausible solutions if any will also add some value to the manuscript.

Author Response
Comment #1
This reviewer believes storage is not very important and may not be required in the title. Only computing and Networking is ok!
Response #1
We thank reviewers for the comments, but the authors believe that storage is as important as computing and networking. Therefore, we keep the “storage-related” discussion, and add more information regarding storage in background (c.f. Section 3.2) and in the workflow (c.f. Section 6.2).
Comment #2
Highlights on the major contributions with emphasis seems missing in the abstract.
Response #2
We have updated our abstract.
Comment #3
Intro 1st para: Machina Research announces its acquisition by Gartner (the world’s leading information technology research and advisory company) long back.
Response #3
We have added information regarding this information (c.f. line 25)
Comment #4
Page-2 Motivations to this work requires substantial improvements with clarity of flows and arguments since it is redesigning the architecture and challenging both the layering centralized approach in some sense. This is definitely a delicate balancing act.
Response #4
We have added background and motivation in Section 3.
Comment #5
Literature Page-3: There are a few important works elaborating the decentralization even in the case of mobile IoT networks. It seems that some relevant and important works are not covered here.
Response #5
We have added more related papers in Section 2.
Comment #6
Section 3.4 requires further elaboration with literature and benchmarking.
Response #6
We have elaborated more details in the new manuscript. Furthermore, to make it consistent with Figure 1, we add root SDN switches and ISP servers’ definition in this section (c.f. Section 4.4).
Comment #7
Overlay networks: It requires strong justification with valuable references from literature to support such ideas.
Response #7
The overlay network contains multiple shared services that IoT entities can use in the architecture. Because there is no such boundary on the what kind of services that one can offer. We limit ourselves by showing three main shared services that are possible to be implemented in our architecture. We obtain those idea from our literature survey. We have edited our overlay networks section to better describe our intention (c.f. Section 5.1).
Comment #8
Storage workflows: This reviewer is not fully convinced with the storage workflows. Perhaps we can move it as a subsection of the network. I know this is indeed a major change that requires authors to polish the manuscript from top to bottom to tone down the storage aspects and highlight networking and computing aspects.
Response #8
We have edited the storage workflow to better draws out the role of P2P Storage Overlay. We also added the benefits of storing IoT data in a decentralized manner using our architecture (c.f. Section 6.2)
Comment #9
Computing workflows: How it works is explained clearly however, its significance and importance, challenges, impeding issues etc have not been covered thoroughly, which is indeed what the readers and the research community will be looking for in such papers.
Response #9
We have added the benefits of the computing workflows using our architecture (c.f. Section 6.1). Similarly, we also added the benefits for networking workflow (c.f. Section 6.3).
Comment #10
Discussion: We need to bring out some limitations, shortcomings and complexity of handing such architecture in future IoT networks. Any supporting preliminary results, if viable to produce helps to justify such development and findings in the paper. BTW, potential future directions and challenges to be tackled with plausible solutions if any will also add some value to the manuscript.
Response #10
We have enhanced our discussion section (c.f. Section 7) to include the limitations of computing, storage, and networking overlay (i.e., blockchain, decentralized storage, and SDN) for IoT use cases. We also elaborated the future direction for each overlay. Finally, we agree that preliminary results will be a good value for our discussion. However, due to bandwidth constraint, we limit our scope to only proposing a concept paper in this manuscript. The future implementation and evaluation can be discussed in the future work.
Reviewer 3 Report
The paper presents a rethinking of the IoT architecture to decentralize it along three aspects - computing, storage, and networking. The paper is mostly well written and presents information in a clear and concise manner and the figures are clear and well explained. However, the reviewer feels that the paper can benefit from some minor revisions to strengthen it.
General comment:
As claimed by the authors, the paper has potential to serve as a reference for future works that build on this relatively nascent field. However, the reviewer found it difficult to point out out the novelty presented by this work in the sections succeeding section 2 (related works). In other words, if prior work has focused on one or two of the aspects, it would be useful if the authors can clearly point out the advantages/differences in the proposed design along those aspects. This is to clarify the following: Is all of this a new idea? or has prior literature proposed subsets of the architecture proposed by the authors? This needs to be elaborated either in the beginning (related works section) or the descriptions of the components in the architecture, with a brief summary at the conclusion.
The writing can also be benefit from close editing in various parts of the paper.
Specific comments:
Line 40: “quickly” instead of fastly is more accurate.
Line 43 - 49: Some of the tenses used seem inappropriate. For example, “begin”, “gains”, “proves”, “brings” which indicate present work need to be corrected since they are ongoing or have already been implemented.
Line 68: Please correct reference citations to include et al. if necessary.
Line 110: correct to “including”
All Figures: The text is clear apart from the legend. Please increase font size in all legends in figures to make the text more legible.
Line 143: Missing “
Line 205: The reviewer recommends changing the acronym “ASes” and “ISPes” in the document to AS’s and ISPs for accuracy.
Line 327: The reviewer recommends changing title to “IoT workflow examples using the proposed Architecture” or something indicating that examples or use-case scenarios follow the subheading.
Discussion: It is good that the authors mention some of the flaws and research directions in the proposed architecture. However, the first part pertains to limitations related to Ethereum blockchain, which was only taken as an example for easy explanation. If possible, it would be beneficial to briefly list/describe any alternatives that can be implemented with this architecture but with less limitations. This ties-in with the abstract which mentions "scalable and responsive" architecture.
With a recent rise in blockchain interest, there has also been a rise in concerns regarding the increased power usage owing to mining and use of multiple nodes for redundancy. If the authors can think of any advantages of their proposed architecture on the power usage or possible optimizations to reduce overall power consumption, it would be a useful addition. A potentially useful article is https://doi.org/10.1007/s12599-020-00656-x .
Line 412: correct to “pose”
Author Response
Comment #1
The paper presents a rethinking of the IoT architecture to decentralize it along three aspects - computing, storage, and networking. The paper is mostly well written and presents information in a clear and concise manner and the figures are clear and well explained. However, the reviewer feels that the paper can benefit from some minor revisions to strengthen it.
Response #1
We thank the reviewers for their useful comments.
Comment #2
General comment:
As claimed by the authors, the paper has potential to serve as a reference for future works that build on this relatively nascent field. However, the reviewer found it difficult to point out out the novelty presented by this work in the sections succeeding section 2 (related works). In other words, if prior work has focused on one or two of the aspects, it would be useful if the authors can clearly point out the advantages/differences in the proposed design along those aspects. This is to clarify the following: Is all of this a new idea? or has prior literature proposed subsets of the architecture proposed by the authors? This needs to be elaborated either in the beginning (related works section) or the descriptions of the components in the architecture, with a brief summary at the conclusion.
The writing can also be benefit from close editing in various parts of the paper.
Response #2
We have added more description of related papers in Section 2, we also rephrase the last paragraph of this section to better describe our difference with the related papers.
Comment #3
Specific comments:
Line 40: “quickly” instead of fastly is more accurate.
Response #3
Fixed (c.f. line 43)
Comment #4
Line 43 - 49: Some of the tenses used seem inappropriate. For example, “begin”, “gains”, “proves”, “brings” which indicate present work need to be corrected since they are ongoing or have already been implemented.
Response #4
Fixed.
Comment #5
Line 68: Please correct reference citations to include et al. if necessary.
Response #5
Re-checked done.
Comment #6
Line 110: correct to “including”
Response #6
Fixed.
Comment #7
All Figures: The text is clear apart from the legend. Please increase font size in all legends in figures to make the text more legible.
Response #7
We have increased the font size of legends across all images in the new manuscript.
Comment #8
Line 143: Missing “
Response #8
Fixed.
Comment #9
Line 205: The reviewer recommends changing the acronym “ASes” and “ISPes” in the document to AS’s and ISPs for accuracy.
Response #9
Fixed.
Comment #10
Line 327: The reviewer recommends changing title to “IoT workflow examples using the proposed Architecture” or something indicating that examples or use-case scenarios follow the subheading.
Response #10
We have changed the section title (c.f. Section 6).
Comment #11
Discussion: It is good that the authors mention some of the flaws and research directions in the proposed architecture. However, the first part pertains to limitations related to Ethereum blockchain, which was only taken as an example for easy explanation. If possible, it would be beneficial to briefly list/describe any alternatives that can be implemented with this architecture but with less limitations. This ties-in with the abstract which mentions "scalable and responsive" architecture.
Response #11
We have improved our discussion section (c.f. Section 7) to include an alternative or possible solutions to the given issues. The future directions are also explained in a more elaborated way in the new manuscript.
Comment #12
With a recent rise in blockchain interest, there has also been a rise in concerns regarding the increased power usage owing to mining and use of multiple nodes for redundancy. If the authors can think of any advantages of their proposed architecture on the power usage or possible optimizations to reduce overall power consumption, it would be a useful addition. A potentially useful article is https://doi.org/10.1007/s12599-020-00656-x .
Response #12
The proposed architecture sits on top of existing blockchain architecture. Thus, we indirectly inherit this high-power consumption issue. Unfortunately, the objective of this paper is to present a concept paper of decentralizing IoT architecture in a general and wide scope, combining three IoT workflows (i.e., computing, storage, and networking). Therefore, we do not solve the PoW high-power consumption issue. It is better to discuss this issue on a separate paper addressing solely on blockchain consensus, like the paper that the reviewer suggest.
Comment #13
Line 412: correct to “pose”
Response #13
Fixed.
Reviewer 4 Report
This paper designs a decentralized overlay IoT architecture for P2P networking, P2P computing, and P2P storage using the smart contract and blockchain. The authors also provide the storage workflow, computing workflow, and network workflow; furthermore, they claim IoT service developers can use the proposed decentralized overlay networks to implement their application.
However, the authors can improve this work by providing the performance evaluation or the optimization compared with previous/related work. Even though the authors propose several designs, it fails to demonstrate that the proposed work is better than other studies or has strength in terms of performance metrics. One of the performance metrics can be scalable, described in the abstraction section; however, this work does not show how scalable the proposed decentralized architecture is.
The problem and purpose mentioned in the abstraction and introduction section are too general to demonstrate; hence, I recommend the authors be concise and brief on the problem and purpose of this work, and it will have a high chance to demonstrate the paper contribution. The specific problems mentioned are the performance limitation of Ethereum (e.g., 15 transactions per second), data consistency stored in decentralized storages, and interoperability/compatibility among SDN infrastructure components.
Author Response
Comment #1
This paper designs a decentralized overlay IoT architecture for P2P networking, P2P computing, and P2P storage using the smart contract and blockchain. The authors also provide the storage workflow, computing workflow, and network workflow; furthermore, they claim IoT service developers can use the proposed decentralized overlay networks to implement their application.
Response #1
We thank the reviewers for their useful comments.
Comment #2
However, the authors can improve this work by providing the performance evaluation or the optimization compared with previous/related work. Even though the authors propose several designs, it fails to demonstrate that the proposed work is better than other studies or has strength in terms of performance metrics. One of the performance metrics can be scalable, described in the abstraction section; however, this work does not show how scalable the proposed decentralized architecture is.
Response #2
We agree with the reviewer that the addition of implementation and some performance measurements will improve the quality of the discussion in this paper. However, due to bandwidth constraint, we limit our scope as a concept paper for now, and leave further implementation and evaluation as future works.
Comment #3
The problem and purpose mentioned in the abstraction and introduction section are too general to demonstrate; hence, I recommend the authors be concise and brief on the problem and purpose of this work, and it will have a high chance to demonstrate the paper contribution. The specific problems mentioned are the performance limitation of Ethereum (e.g., 15 transactions per second), data consistency stored in decentralized storages, and interoperability/compatibility among SDN infrastructure components.
Response #3
We have improved our Introduction (c.f. Section 1). We also add background and motivation in Section 3.
Reviewer 5 Report
This concept paper proposes a decentralizing architecture for IoT. The authors focus on three aspects of the proposed architecture that are computing, storage, and networking. They pointed out that their architecture divided into low-level and high-level. This study is properly prepared but requires some improvement. The authors must address all of the below concerns carefully.
- We suggest converting the title of the concept paper into a question form.
- In the abstract, the authors pointed out that “we have adopted centralized architecture for decades.” (page1-line1). This sentence is not clear and inaccurate. Is the security issue connected only with computing and storage (page1-line7-8)? The abstract requires some improvement.
- Introduction Section: The second paragraph in the introduction contains no reference. The introduction needs to add references in various places. We recommend that authors add the research question or questions precisely in addition to the research objectives. Where we must find solutions to these questions and achieve goals in the end.
- Related works: The authors presented the related research well, but they did not point out the shortcomings of these related papers. If all of these papers are good and do not contain problems then what is the purpose of creating this research (although the authors indicated two sentences at the end of the section but this is not sufficient).
- Some acronyms should be defined before they can be used, such as "SDN", ... etc. Authors should replace “ASes” with “ASs", “ISPes” with “ISPs” (check the entire concept paper).
- What is Ethereum and why was it chosen to secure data storage and computation? we know what Ethereum is, but the research should be comprehensive when read by researchers, the same of questions for IPFS and OpenFlow.
- Can the performance efficiency of the proposed architecture (Figure 5) be applicable?
- In the discussion, the authors did not indicate that their solutions are applicable to the objectives of this research. The discussion section requires improvement.
- Conclusion Section, we did not find the conclusions for this study. The conclusion section should be improved.
- Figures and Tables: All figures and tables are clear and drawn with high resolution. Nonetheless, Figures 3, 4, 5 and Table1 are shown before being summoned in-text. In Table 1, the year, method, and limitation should be added. Also, it is preferable to add more recent research (2020 and 2021) to Table 1.
- References list: References should follow the MDPI-IoT style. The references list requires minor scrutiny by the authors. Some acronyms in research names require that they be uppercase, such as [10], [11], … etc. The first three references are recent, and this is good. However, it is preferable to replace them with journal recent papers to make the list of references more robust.
- Proofreading: English writing for this search is good. However, this concept paper requires minor proofreading. For example, authors should add “a”, “an” or “the” for “low-level” (page2-line63), “Dragon” (page2-line70) … etc., replace “is” with “are” (page6-line204), use “necessary” instead of “neccesary” (page7), “Aggrement” instead of “Aggrement” (page8-line292), … etc. (check the entire paper).
Author Response
Comment #1
This concept paper proposes a decentralizing architecture for IoT. The authors focus on three aspects of the proposed architecture that are computing, storage, and networking. They pointed out that their architecture divided into low-level and high-level. This study is properly prepared but requires some improvement. The authors must address all of the below concerns carefully.
Response #1
We thank the reviewers for their useful comments.
Comment #2
We suggest converting the title of the concept paper into a question form.
Response #2
We changed the title.
Comment #3
In the abstract, the authors pointed out that “we have adopted centralized architecture for decades.” (page1-line1). This sentence is not clear and inaccurate. Is the security issue connected only with computing and storage (page1-line7-8)? The abstract requires some improvement.
Response #3
We updated the abstract.
Comment #4
Introduction Section: The second paragraph in the introduction contains no reference. The introduction needs to add references in various places. We recommend that authors add the research question or questions precisely in addition to the research objectives. Where we must find solutions to these questions and achieve goals in the end.
Response #4
Fixed.
Comment #5
Related works: The authors presented the related research well, but they did not point out the shortcomings of these related papers. If all of these papers are good and do not contain problems then what is the purpose of creating this research (although the authors indicated two sentences at the end of the section but this is not sufficient).
Response #5
We have added more related papers in Section 2. We also rephrased the last paragraph of this section.
Comment #6
Some acronyms should be defined before they can be used, such as "SDN", ... etc. Authors should replace “ASes” with “ASs", “ISPes” with “ISPs” (check the entire concept paper).
Response #6
Fixed.
Comment #7
What is Ethereum and why was it chosen to secure data storage and computation? we know what Ethereum is, but the research should be comprehensive when read by researchers, the same of questions for IPFS and OpenFlow.
Response #7
We have added this description in Section 3.
Comment #8
Can the performance efficiency of the proposed architecture (Figure 5) be applicable?
Response #8
We agree with the reviewer that the addition of implementation and some performance measurements will improve the quality of the discussion in this paper. However, due to bandwidth constraint, we limit our scope as a concept paper for now, and leave further implementation and evaluation as future works.
Comment #9
In the discussion, the authors did not indicate that their solutions are applicable to the objectives of this research. The discussion section requires improvement.
Response #9
We have updated our discussion section (c.f. Section 7).
Comment #10
Conclusion Section, we did not find the conclusions for this study. The conclusion section should be improved.
Response #10
We have rephrased the conclusion (c.f. Section 8).
Comment #11
Figures and Tables: All figures and tables are clear and drawn with high resolution. Nonetheless, Figures 3, 4, 5 and Table1 are shown before being summoned in-text. In Table 1, the year, method, and limitation should be added. Also, it is preferable to add more recent research (2020 and 2021) to Table 1.
Response #11
We have changed the positioning of figures and tables.
Comment #12
References list: References should follow the MDPI-IoT style. The references list requires minor scrutiny by the authors. Some acronyms in research names require that they be uppercase, such as [10], [11], … etc. The first three references are recent, and this is good. However, it is preferable to replace them with journal recent papers to make the list of references more robust.
Response #12
Fixed for [10] and [11]. As of the webpage references, it is hard to find journals or papers that contains up-to-date statistics of IoT growth.
Comment #13
Proofreading: English writing for this search is good. However, this concept paper requires minor proofreading. For example, authors should add “a”, “an” or “the” for “low-level” (page2-line63), “Dragon” (page2-line70) … etc., replace “is” with “are” (page6-line204), use “necessary” instead of “neccesary” (page7), “Aggrement” instead of “Aggrement” (page8-line292), … etc. (check the entire paper).
Response #13
We have checked the grammar in the new manuscript.
Round 2
Reviewer 4 Report
The authors provide a further description of related work and preliminaries; again, background information and modeling have been well organized and described since the first version.
However, it isn't easy to evaluate the work if authors don't evaluate their work.
Reviewer 5 Report
The authors responded to most of the concerns. However, the authors should address the below comment.
- Introduction Section: The second paragraph in the introduction contains [?] reference. The introduction needs to add references in various places. We recommend that authors add the research question or questions precisely in addition to the research objectives. Where we must find solutions to these questions and achieve goals in the end.